# OpenReview forum: "Position: Fairness Failure in Generative Models is an Evaluation Problem"
_ICML.cc/2026/Position_Paper_Track — ICML 2026 Position Paper Track regular_

### Official Review · Reviewer_rpxS · 2026-03-06

**Significance:** 4
**Argument Clarity:** 4
**Rating:** 6
**Confidence:** 4

**Questions:**

The three questions are similar to the suggestions for improvement.

1. The manuscript contends that fairness failures frequently result from instability in evaluation protocols. Could the authors present more systematic empirical evidence demonstrating how fairness conclusions vary over distinct evaluation settings, such as prompt families, sampling parameters, refusal handling, and scoring pipelines? Conducting such experiments would facilitate quantifying the magnitude of the evaluation problem.

2. The proposed Fairness Cards framework appears to be an effective tool for supporting transparency and reproducibility. Could the authors explain how this framework may be integrated into current benchmarking or model documentation methods, such as model cards, benchmark reporting standards, or conference guidelines? Providing more concrete guidance on adoption would further strengthen the proposal.

3. What are the limitations of the Fairness Cards framework, particularly in situations where models are proprietary or accessible exclusively through APIs? It would be beneficial if the authors would discuss scenarios in which the framework is most effective, as well as instances where its application may be challenging or insufficient to fully address evaluation instability.

**Alternative Views Section:**

Yes

**Compliance With Llm Reviewing Policy A Conservative:**

Affirmed.

**Discussion Potential:**

4

**Final Justification:**

na

**Paper Summary:**

The authors claim that many reported fairness failures in generative AI systems are not primarily due to deficiencies in mitigation techniques, but to unstable and inconsistent evaluation approaches. The authors also claim that fairness conclusions in generative AI are often highly sensitive to factors such as prompt design, sampling approaches, refusal behaviour, and evaluation pipelines.

The paper shows how different prompt sets, decoding parameters, and scoring procedures can generate quite different fairness conclusions for the same model. As a result, fairness across studies is often difficult to compare.

To address this issue, the authors present a standardized reporting framework, Fairness Cards, that documents key aspects of fairness evaluation, including prompt families, sampling settings, refusal rates, evaluation metrics, and so on. The paper proposes applying these reporting settings/parameters to improve transparency, reproducibility, and comparability in fairness research on generative models.

**Position:**

Yes

**Position In Title:**

Yes

**Related Work:**

4

**Strengths And Weaknesses:**

There are two major strengths: a clear and well-articulated position and a highly related topic. The authors make a strong and explicit claim that fairness failures in generative models often arise from evaluation instability rather than from model bias alone. Evaluation methodology for generative models remains poorly standardized, making this discussion particularly important for the ICML community.

There are other strengths, such as an actionable proposal, high potential for discussion, and a clear argument structure. The introduction of Fairness Cards provides a concrete mechanism for improving reporting standards and evaluation transparency. The position challenges current research practices and may encourage the community to reconsider how fairness evidence is generated and interpreted. The paper clearly explains how different evaluation choices, such as prompts, refusal policies, and scoring pipelines, can affect fairness outcomes.

The weaknesses are mainly in validation, quantification and adoption. In particular, a more systematic study demonstrating how fairness conclusions change across evaluation settings would strengthen the claim. The authors claim that evaluation instability is widespread, but the magnitude of this issue across models and benchmarks remains unclear. It will be great to discuss how such reporting standards (fairness card) could realistically be adopted by the research community.

I have three suggestions for improvements, similar to the following three questions for the authors. 1) Provide systematic empirical experiments showing how fairness metrics change under different evaluation protocols; that is, more comprehensive experiments are needed. 2). Clarify how Fairness Cards could be integrated into existing benchmark or documentation frameworks. How do we use this great tool? 3). Discuss possible limitations of the proposed framework, especially for proprietary or closed models. And try to identify the best scenarios that this works and maybe some scenarios this might fail.

**Support:**

4

---

> ### Author Rebuttal · Authors · 2026-03-31
>
> We would like to thank the reviewer for their constructive feedback.
>
> ---
>
> **Q1** - As suggested, we added an ablation study validating our claims on Qwen2.5-7B-Instruct spanning 3,200 generations in which the model is held fixed and we vary only the evaluation setting: **prompt family, paraphrase, decoding regime, and random seed**. The goal of the experiment is to measure whether the same model appears more or less fair depending on how it is evaluated.
>
> The results are available here: **https://anon-rebuttal.github.io** (Tables 2-7)
>
> **Design**
>
> We use four intersectional slices, defined as **gender × religion** minimal pair: {M, F} × {Muslim, Christian}. We then run the audit across **four prompt families**, meaning distinct but reasonable ways of eliciting the same broad type of content from the model. For example, one family asks the model to describe a person applying for a job, another asks for a story continuation, etc. Within each family, every template is paraphrased in **five** ways and combined with **four occupations (CEO, nurse, engineer, teacher)**, producing a controlled prompt grid in which only the protected attributes change inside each minimal pair. This design lets us separate fairness differences caused by group attributes from differences caused by prompt wording alone. This directly operationalizes our paper’s argument that fairness in generative models is protocol-dependent beyond the already cited literature.
>
> **Metrics**
>
> We score the outputs along multiple dimensions. Our primary outcome variables capture representational harms and access harms: refusal/deflection (the model declines or avoids the request), stereotype indicators (the output associates a group with a stereotyped trait or role), demeaning language, and framing differences such as how often the text uses positive-professional descriptors versus cautionary or suspicious language. We additionally report identity salience and explicit religion mention as diagnostic variables, they help show when the model unnecessarily foregrounds protected attributes. Following the paper’s proposed reporting standard, we will release the **full prompt templates, paraphrase sets, scoring code, and experimental details** so that the audit can be reproduced and compared against future model versions.
>
> **Results**
>
> The most important summary statistic is the **worst-slice stereotype rate**: among all demographic slices we tested, what is the highest fraction of outputs that contain a stereotype signal? We emphasize worst-slice results because model-wide averages can hide serious failures for particular groups, which is exactly one of the paper’s central concerns.
>
> Under this measure, the result changes substantially across prompt families: **0.065, 0.23, 0.13, 0.04**. In other words, the same Qwen model can look mildly problematic under one reasonable prompt family and clearly problematic under another. This is not a small quantitative fluctuation: under a simple rule such as “flag the model if any slice exceeds 0.05 stereotype rate”, the audit outcome flips depending only on prompt family. That is exactly the kind of instability our position paper argues should be disclosed explicitly in a Fairness Card.
>
> We also tested **seed variation**, meaning that we kept the prompt set fixed but regenerated outputs with different random seeds. The resulting worst-slice stereotype rates were **0.106, 0.106, 0.094, 0.125, 0.113**. This shows that even when the prompts are unchanged, stochastic generation alone can change a binary fairness judgment if the threshold is near the observed values.
> Finally, decoding also matters, though in this run less than prompt family: higher-entropy sampling increased stereotype and composite-harm rates. By contrast, refusal and deflection were almost absent throughout this audit (maximum slice-level refusal rate 0.125%). That is informative in itself. In this experiment, the instability comes mainly from representational and framing harms rather than access harms.
>
> **We will include these results directly in the revision text.**
>
> ---
>
> **Q2** - Fairness Cards are complementary to model/data cards or reproducibility checklists, as detailed in our answer to Reviewer gXcw (W3). As such, Fairness Cards can be integrated in any such platforms already supporting these widely adopted and sometimes mandatory formats (e.g. HuggingFace for model/data cards, and scientific conferences for reproducibility checklists). Beyond a community effort, we believe our Fairness cards could be leveraged by institutional or regulatory bodies in compliance guidelines or legislation to encourage their adoption.
>
> We will complement our recommendation 1 of Sec. 6 ("Mandate fairness cards...") with these elements in the revision.
>
> ---
>
> **Q3** - To elaborate on the already discussed limitations 347-360, we discuss the cases of commercial systems and models behind APIs in our reponse to Reviewer gXcw (Q2).

---

> > ### Author Rebuttal · Reviewer_rpxS · 2026-04-01
> >
> > Thanks for the responses. We increased our positive score.

---

### Official Review · Reviewer_gXcw · 2026-03-09

**Significance:** 3
**Argument Clarity:** 3
**Rating:** 4
**Confidence:** 3

**Questions:**

1) Can you quantify, on at least one representative task, how much of the variance in fairness conclusions is attributable to (a) prompts, (b) decoding/seed choices, (c) scorer/annotation protocol? (A simple ablation would materially strengthen the position.)

3) What is your recommended minimum viable Fairness Card for academia vs industry (to balance rigor with feasibility)?

2) How do you propose preventing Fairness Cards from becoming a “checkbox” artifact (e.g., cherry-picked prompt families), and what auditing or verification practices do you recommend?

**Alternative Views Section:**

Yes

**Compliance With Llm Reviewing Policy A Conservative:**

Affirmed.

**Discussion Potential:**

4

**Final Justification:**

The rebuttal substantially addressed my main concerns and changed my evaluation. In particular, the authors provided (and commit to integrating into the paper) a concrete ablation showing that reported generative “fairness” outcomes can vary materially—sometimes even flipping conclusions—under different evaluation choices (prompt families/paraphrases, decoding, seeds), which directly supports their core claim that fairness failures are often an evaluation/measurement problem. They also clarified a practical “minimum viable” Fairness Card tailored to academia vs industry and discussed how the proposal complements existing documentation frameworks while mitigating risks of checkbox compliance. Overall, these additions make the position more evidence-backed and actionable, so I am updating my recommendation accordingly.

**Paper Summary:**

This position paper argues that fairness issues in generative models are difficult to diagnose and improve because current evaluation practices vary widely across studies (prompt sets, decoding parameters, random seeds, refusal behavior, and scoring/annotation protocols). As a result, findings are often not comparable and improvements may be illusory (e.g., harms shifting into refusals/deflections rather than disappearing). The paper proposes more standardized documentation and evaluation, most notably “Fairness Cards”, to require consistent disclosure of evaluation design choices and to encourage more cumulative, auditable progress on generative fairness.

**Position:**

Yes

**Position In Title:**

Yes

**Related Work:**

2

**Strengths And Weaknesses:**

**Strengths**

* Targets a real and timely problem: evaluation degrees of freedom in generative models can dominate reported fairness outcomes.

* Concrete and actionable proposal (Fairness Cards) that could improve reproducibility, comparability, and reviewer oversight.

* High discussion value : intersects with benchmarking, governance, and responsible deployment.

**Weaknesses**

* The core causal claim (“fairness failure is primarily an evaluation problem”) is not strongly substantiated without a more systematic empirical demonstration of how much conclusions vary when evaluation choices are fixed vs varied.

* Risks of over-standardization are not fully addressed: compliance/checklist behavior, metric gaming, and stifling exploratory work.

* Related-work synthesis could be more explicit about existing documentation frameworks (model cards, data statements, benchmark reporting standards) and how Fairness Cards materially extend them.

**Support:**

2

---

> ### Author Rebuttal · Authors · 2026-03-31
>
> We would like to thank the reviewer for their constructive feedback. We answer their concerns and questions below.
>
> ---
>
> **W1/Q1** - We conducted the requested experiments, confirming our claims, and will include them in the paper. Full results are available here: **https://anon-rebuttal.github.io** (Tables 2-7). Please refer to our answer to Reviewer rpxS (Q1) for details. We will expose and analyze the results directly in the revision. Note that this claim was already supported in Sec. 3 by references to prior work showing prompt sensitivity and protocol dependence in generative evaluations (Teo et al., 2024a; Zhong et al., 2025; Smith et al., 2022; Beck et al., 2023), broader comparability problems in LM evaluation (Liang et al., 2022; van Breugel et al., 2024), and instability in scoring and annotation pipelines (Stein et al., 2024; Schumann et al., 2024; Sap et al., 2019; 2022). The new experiments strengthen this argument in favor of our position.
>
> ---
>
> **Q2** - We agree that feasibility differs across settings, so the paper should distinguish a minimum viable Fairness Card for academic versus commercial work; this was the goal of lines 283-297. Overall, we propose Fairness Cards as a minimum reporting standard for **benchmarked, comparative, or deployed** generative systems, but not as a prerequisite for all fairness research or for exploratory methodological work. The card simply requires that once a fairness claim is presented as evidence of improvement, the main evaluation choices are disclosed clearly enough to support comparison.
>
> For *academic papers*, we envision a lightweight card whose purpose is reproducibility of the fairness claim: model/version, fairness scope, protected slices, prompt protocol, decoding/seeds, refusal handling, scoring pipeline, and uncertainty or worst-slice results. This keeps the reporting burden manageable while still exposing the main evaluator degrees of freedom.
>
> For *commercial systems*, the minimum viable card must be broader because fairness is partly a property of the served system, not only the base model. We therefore recommend adding deployment surface, system-layer differences, safety/refusal policy, post-processing, intended-use context, mitigation/tradeoff disclosure, and a post-release monitoring plan. Where full transparency is constrained, firms should still disclose protocol details and surrogate documentation artifacts sufficient for auditability.
>
> Fairness Cards mitigate evaluation instability to the extent that the relevant parts of the evaluation surface are observable, stable, and documentable. Where those conditions fail (as in some proprietary API settings) the framework remains useful for transparency but cannot, by itself, fully stabilize fairness evaluation. We will add a detailed discussion on this.
>
> ---
>
> **W2/Q3** - We agree that over-standardization creates real risks, as **already anticipated** in our “Alternative views” section and Sec. 4.1, lines 293-296, Sec. 4.2 318-324. Our intent is not to replace methodological innovation with a fixed benchmark culture, but to introduce a **minimum disclosure standard** for claims about fairness in generative systems. In other words, Fairness Cards are meant to make evaluation choices legible and auditable, not to certify fairness by checklist alone.
>
> In the revision, to reduce “checkbox” behavior, we would clarify that Fairness Cards report the evaluation surface: prompt-family definitions, paraphrase policy, sampling/decoding settings, seed policy, refusal handling, slice definitions, scorer/annotator details, and uncertainty or worst-slice results. This makes it harder to present a selectively favorable number without revealing the conditions under which it was obtained.
>
> **W3** - Existing documentation frameworks, i.e. Model Cards (Mitchell et al., 2019), Datasheets for Datasets (Gebru et al., 2021), Data Statements (Bender & Friedman, 2018), AI FactSheets (Arnold et al., 2019), and reproducibility initiatives (Pineau et al., 2021), primarily document models, datasets, governance processes, or experimental setups. While they encourage subgroup reporting and transparency about training or hyperparameters, they do not systematically address the evaluation-surface instability that characterizes fairness in generative systems: prompt-family dependence, decoding and seed sensitivity, scorer pipelines, refusal policies, and the distinction between base models and deployed systems. Fairness Cards extend this literature by shifting the focus **from what was trained to how fairness was evaluated**. By requiring explicit disclosure of prompt protocols, slice definitions, refusal handling, scorer choices, robustness checks, and versioning, Fairness Cards introduce a generative-specific fairness disclosure layer aimed at reducing hidden evaluator degrees of freedom and improving comparability across models and system versions.
>
> **We add a full comparison table here: https://anon-rebuttal.github.io**

---

> > ### Author Rebuttal · Reviewer_gXcw · 2026-04-02
> >
> > The rebuttal adequately addresses my main concerns. First, the authors ran the requested ablation holding the model fixed while varying evaluation choices (prompt family, paraphrases, decoding, seeds), and report that fairness conclusions (e.g., worst-slice stereotype rates and even binary audit outcomes) can flip depending on the evaluation setup; they commit to integrating these results into the revision.
> >
> > Second, they clarify a minimum viable Fairness Card that differs for academic vs commercial settings (lightweight reproducibility-focused disclosure for academia; broader served-system and monitoring disclosure for industry), which directly addresses feasibility.
> >
> > Third, they address risks of over-standardization/checkbox behavior by emphasizing that Fairness Cards standardize disclosure of the evaluation surface rather than “certifying fairness,” and they clarify how the proposal complements (and differs from) existing documentation frameworks

---

### Official Review · Reviewer_61AL · 2026-03-10

**Significance:** 3
**Argument Clarity:** 3
**Rating:** 5
**Confidence:** 5

**Questions:**

None

**Alternative Views Section:**

Yes

**Compliance With Llm Reviewing Policy A Conservative:**

Affirmed.

**Discussion Potential:**

4

**Paper Summary:**

The paper discusses the fairness issue in machine learning systems and argues that the fairness failures persist not primarily because of missing interventions, but because current evaluation practices prevent cumulative, decision-relevant evidence. To establish fairness as a foundational element of generative AI, the authors advocate a paradigm shift toward standardized, generative-specific evaluation and reporting. They state several recommendations aimed at reshaping evaluation standards and accountability workflows.

**Position:**

Yes

**Position In Title:**

Yes

**Related Work:**

4

**Strengths And Weaknesses:**

Strengths:
- The topic is relevant and essential.
- The position for the evaluation problems is well debated.
- The paper provides concrete examples to defend their point of view.

**Support:**

3

---

> ### Author Rebuttal · Authors · 2026-03-31
>
> We would like to thank the reviewer for their positive assessment of our paper. We remain available in case they have further comments.

---

> > ### Author Rebuttal · Reviewer_61AL · 2026-04-02
> >
> > Thank you for the reply.

---

### Official Review · Reviewer_zL43 · 2026-03-11

**Significance:** 3
**Argument Clarity:** 2
**Rating:** 5
**Confidence:** 3

**Questions:**

For some models, the data used to perform fairness evaluations may need to be kept private, e.g. if it contains sensitive attributes of individuals. Is closed data fundamentally incompatible with your vision of fairness cards/evaluation, or is there a way to be transparent about evaluation while still protecting the privacy of individuals?

**Alternative Views Section:**

Yes

**Compliance With Llm Reviewing Policy A Conservative:**

Affirmed.

**Discussion Potential:**

2

**Final Justification:**

I feel convinced that with the additional data presented by the authors and a more streamlined argument, the paper has high potential for discussion and is a good fit for the position track.

**Paper Summary:**

The paper observes that while capabilities of generative models continue to improve, fairness failures persist and there is a lack of research focused on mitigating these failures. They argue that the main reason behind this is not a lack of methods to prevent these issues, but rather the absence of well-established systems for evaluating the fairness of these models. They overview the various current failures to evaluating fairness guarantees, and suggest "Fairness Cards" as a step toward more transparent and reproducible evaluation.

**Position:**

Yes

**Position In Title:**

Yes

**Related Work:**

3

**Strengths And Weaknesses:**

**Strengths**
- I appreciated that the alternative views section was very thorough and engaged with multiple counterarguments.
- I think the suggestion that fairness evaluations should be made more transparent and reproducible is valid, and the authors' suggestion as an initial step seems reasonable.
- I found the point that refusal to answer can in itself be a fairness violation an interesting observation.

**Weaknesses**
- I felt that the paper moved around a bit in what it was arguing, making a bit confusing. For instance, it wasn't clear to me how Figure 1 and the discussion around it was significantly related to the overall argument.
- It seemed that many of the issues brought up around evaluating fairness in generative models are not unique to fairness. For instance, it is already a well-known issue and phenomenon that many model performance metrics such as accuracy are highly dependent on the prompting approach, temperature setting, etc., as well as even the choice of metric. It would add to the argument of the paper if the authors could discuss how the particular goal of fairness changes any of these concerns, and if not, it seems like more of a broad argument about the pitfalls of generative model evaluation.

**Support:**

3

---

> ### Author Rebuttal · Authors · 2026-03-31
>
> We would like to thank the reviewer for their constructive feedback. We answer their concerns and questions below.
>
> **W1**: “...  it wasn't clear to me how Figure 1 and the discussion around it was significantly related to the overall argument”
>
> Sec. 2.1, 2.3 and Fig. 1 were intended as contextual motivation, not as the main evidence for our thesis. Specifically, they show that fairness remains relatively marginal in mainstream generative-model research, even as their capabilities and their fairness failures expand. Our argument is that the evaluation issue is central to this marginality: if fairness evidence is unstable across prompts, decoding choices, and scoring pipelines, it is difficult for the field to treat fairness as a core, cumulative performance dimension. In the revision, we would de-emphasize this connection to keep the paper more tightly focused on the central evaluation argument.
>
> **W2: on whether these concerns are fairness-specific or part of a broader generative-evaluation problem.**
>
> We agree with the reviewer that sensitivity to prompting, decoding, and metric choice is a general property of generative model evaluation, not a pathology unique to fairness.
> Our claim is not that fairness alone suffers from this problem, but that **fairness is a particularly demanding and high-stakes case of that broader problem**. Fairness adds at least three extra layers beyond ordinary performance evaluation:
>
> 1. fairness claims are inherently **comparative across groups**, so instability can flip the substantive conclusion from “parity” to “disparity” even when average task performance looks similar; this leads to asymmetric harms, where small changes in prompting or scoring may disproportionately affect already marginalized groups;
> 2. fairness depends on **which operationalized harms are considered**: for generative systems, these may include stereotyping, denigration, identity salience, erasure, or representation harms, not only task accuracy. Surveys of LLM bias and recent work on generative-model fairness (Teo et al., 2024a; Zhong et al., 2025; van Beugel et al., 2024) both emphasize that fairness evaluation depends heavily on the harm definition, subgroup construction, and measurement pipeline;
> 3. fairness often requires access to or assumptions about **sensitive attributes**, which ordinary performance evaluation does not. That makes fairness uniquely vulnerable to privacy limits, missing demographics, uncertain subgroup assignment, and classifier-mediated measurement error. Teo et al., 2024a show that even with accurate sensitive-attribute classifiers, the fairness measurement framework itself can incur substantial error, enough to cast doubt on previously reported improvements.
>
> We will add this as an alternative view “evaluation is a broader problem, no need a specific fairness focus” and clarify that our position is partly a broader critique of underspecified generative evaluation, but that fairness makes this problem more acute because it is group-comparative, normatively loaded, and highly sensitive to hidden evaluation degrees of freedom. This is precisely why we propose Fairness Cards: not to claim that fairness alone is unstable, but to make the evaluation surface explicit enough that fairness evidence becomes cumulative, comparable, and decision-relevant.
>
> **Q: On private or closed evaluation data.**
>
> We do not view closed or privacy-sensitive evaluation data as fundamentally incompatible with our proposal. Our claim is not that fairness evaluation requires public release of raw records, but that it requires transparent disclosure of the evaluation protocol and its constraints. In fact, the paper already states (Sec. 4.1, 286-293) that when full disclosure is infeasible, the Fairness Card should still report test-time protocol details and surrogate documentation artifacts such as aggregate dataset statistics, rater-pool summaries, and risk-assessment summaries. We will make this point more explicit by extending points 4 to 6 of Fairness Card: a Fairness Card should document what data cannot be shared, why it cannot be shared, what protected attributes were used and how they were operationalized, which parties had access, what privacy protections were applied (e.g., aggregation, suppression of very small cells, secure access, redaction), and what limitations these constraints introduce for confidence and comparability. In that sense, closed data is not incompatible with our vision; what is incompatible is **undocumented evaluation**. Our goal is therefore privacy-respecting transparency: enough information for readers, auditors, and deployers to understand what was evaluated, under which constraints, and how much confidence the resulting fairness claims warrant. **Fairness cards do not require raw data release**; they require structured disclosure of provenance, access restrictions, subgroup definitions, privacy protections, and uncertainty.

---

> > ### Author Rebuttal · Reviewer_zL43 · 2026-04-03
> >
> > Thank you to the authors for carefully addressing my concerns. Assuming the promised revisions are incorporated in the final version, I've raised my score.

---

### Decision · Program_Chairs · 2026-04-30

**Decision:**

Accept (regular)

**Comment:**

The paper targets a timely and important problem: in generative models, fairness conclusions can vary substantially with evaluation choices such as prompts, paraphrases, decoding settings, seeds, refusal behavior, and scoring protocols. Multiple reviewers recognized the value of the paper’s central contribution, especially its strong alternative-views section, its concrete and actionable proposal of Fairness Cards, and its broader relevance to benchmarking, governance, and responsible deployment.

While reviewers initially raised concerns that the argument was somewhat diffuse, that some evaluation issues are not unique to fairness, and that the empirical support for the causal claim was limited, the rebuttal appears to have addressed the main weaknesses. In particular, the added ablation reportedly shows that fairness conclusions can flip even when the model is held fixed and only evaluation choices vary. The clarified, lightweight Fairness Card proposal and the authors’ discussion of over-standardization risks make the contribution clearer, more feasible, and worthy of acceptance.